# Quality of life, self-reported outcomes and impact of education among people with moderate and severe hemophilia A: An integrated perspective from a Latin American country

Liliana Torres[1], Oscar Peñuela[1], Maria del Rosario Forero[1], Juan Satizabal[2], Ximena Salazar[2], Diana Benavides[2], Raul Gamarra[2], Marcela Rivera[3], David Vizcaya[3], Juan-Sebastian Franco[1] *

1 Medical Affairs, Bayer S.A., Bogotá, Colombia, 2 IPSOS Napoleon Franco S.A., Bogotá, Colombia, 3 Bayer Pharmaceuticals, Sant Joan Despi, Spain

* juansebastian.franco@bayer.com

## Abstract

Collecting and interpreting self-reported outcomes among people with hemophilia A supports the understanding of the burden of the disease and its treatment to improve holistic care. However, in Colombia, this information is limited. Therefore, this study aimed to describe the knowledge, perception and burden of hemophilia A from the patients' perspective. A cross-sectional study was conducted in the context of a hemophilia educational bootcamp held from November 29th to December 1st, 2019, in Medellin, Colombia. The bootcamp was organized by a hemophilia patient association responsible for contacting and inviting patients with hemophilia A (PwHA). Information on patients' health beliefs, treatment experiences, and health-related quality of life (HRQoL) was obtained through focus groups, individual interviews and the Patient Reported Outcomes, Burdens and Experiences (PROBE) questionnaire. A total of 25 moderate or severe PwHA were enrolled in this study and completed the PROBE questionnaire. Acute pain was the most frequently reported symptom, with 88% of the patients reporting the use of pain medication. Difficulty with activities of daily living was reported by 48%. Furthermore, 52% reported having more than 2 spontaneous bleeding events in the last year. Treatment was administered at home for 72% of patients, with regular prophylaxis as the most common treatment regimen. In terms of overall HRQoL, the median EQ-5D VAS score was 80 (IQR: 50–100). PwHA in Colombia still suffer from disease complications related to bleeding events, pain and disability that affect their HRQoL, which highlights the need to develop patient-centered initiatives to improve the wellness of this population.

**Data Availability Statement:** All relevant data are within the manuscript and its Supporting Information files.

**Funding:** The study was funded by Bayer SA Colombia. Medical writing was provided by Ana Maria Perez and editorial assistance was provided by American Journal Experts, North Carolina, USA, funded by Bayer SA Colombia. This study was conducted in accordance with Good Pharmacoepidemiology Practices. The study protocol was approved by an expert committee from Bayer prior to study initiation and the analysis dataset is available as supplementary material.

**Competing interests:** Juan Satizabal, Ximena Salazar, Diana Benavides and Raul Gamarra are employees of IPSOS Napoleon Franco S.A. Liliana Torres, Oscar Peñuela, Maria del Rosario Forero, Marcela Rivera, David Vizcaya and Juan-Sebastian Franco are employees of Bayer. All authors have no further conflicts to disclose. This does not alter our adherence to PLOS ONE policies on sharing data and materials.

# Introduction

Hemophilia A is a rare hereditary bleeding disorder caused by a deficiency of coagulation factor VIII (FVIII) and characterized by repeated and prolonged bleeding into muscles and joints that lead to pain, physical limitations and a negative impact on health-related quality of life (HRQoL) [1]. Prophylaxis is the preferred treatment for patients with severe hemophilia A (PwHA), given its positive impact on joint bleeding events, arthropathy, absenteeism, physical health, pain and HRQoL [2, 3]. However, a considerable number of PwHA have suboptimal adherence to treatment, increasing the probability of bleeding events and long-term disability [1, 4]. Several barriers to adherence are related to the patient condition, assigned treatment, health care system, and socioeconomic aspects, especially in countries that suffer from large social inequalities, such as Latin American countries [5]. In addition, health beliefs around the disease and treatment have been associated with the patient's experience, influencing their understanding of the diagnosis, the prognosis, and the potential benefits of prophylaxis treatment [1, 6], reinforcing the need for personalized treatment strategies for prophylaxis therapy [7]. In this regard, disease-specific bootcamps act as educational initiatives for PwHA to allow them to speak openly, ask questions, confide and learn factual information about this disorder [8, 9].

As with other diseases, collecting and evaluating patient-reported outcomes (PROs) allows a better understanding of perspectives from PwHA on their disease burden and its treatment and is a suitable tool to achieve holistic care. Furthermore, the combination of data collected via PROs with qualitative data collection through interviews and focus groups allows for capturing more meaningful insights into patients' beliefs and perceptions [10]. Recent publications have shown considerable improvement in enhancing content validity in PRO instruments as well as implementation strategies by incorporating the patient's voice through qualitative methods [11, 12]. Despite substantial use of mixed methods for developing or enhancing measurement tools in hemophilia, there is no prior use of such a mixed approach for a holistic quantitative and qualitative clinical patient-centered evaluation within a single PwHA cohort.

According to government data, in 2021, the prevalence of hemophilia A in Colombia was 4.29 cases per 100,000 people, which corresponds to approximately 2,100 people currently living with the condition [13]. Unfortunately, there is limited information on perceptions and experiences among PwHA in Colombia and their potential impact on treatment adherence, health-related quality of life (HRQoL) and clinical outcomes, information that is critical to developing patient-centered initiatives. To our knowledge, only three recent studies have evaluated HRQoL in a similar population with hemophilia from Colombia [14–16]. These studies reported self-perceived HRQoL similar to that in the general population and are consistent with other studies in Latin America [17]. None of these studies used the Patient Reported Outcomes, Burdens and Experiences (PROBE) questionnaire, however. Likewise, no prior study in Latin America incorporated a mixed qualitative and quantitative methodology to evaluate the patient experience in a holistic manner. The aim of the present study was to explore the health beliefs, treatment experiences, HRQoL and impact of educational activities on disease knowledge among PwHA in the context of a hemophilia educational bootcamp.

# Materials and methods

## Study design and participants

This was a cross-sectional observational study conducted in the context of a hemophilia educational bootcamp held from November 29th to December 1st, 2019, in Medellin, Colombia.

The bootcamp included 30 PwHA whose ages ranged from 10 to 59 years. Patients aged <18 were accompanied by one of their parents or legal guardians. This 3-day camp was designed to encourage participants to meet other PwHA, share experiences and increase knowledge of the disease. The bootcamp was coordinated by a hemophilia patient organization (*Liga Antioqueña de Hemofilia*) and sponsored by Bayer. The hemophilia patient organization was responsible for contacting and inviting PwHA to the bootcamp, following local regulations. During the first day, to conduct bonding activities, the participants were randomly distributed into three different groups. On the second day, educational activities were performed, as well as recreational activities. On the third and last day, qualitative interviews and closure of the camp took place. The activities were guided by three counsellors: two social workers and a nurse. Additionally, the campsite had two paramedics and an ambulance available at any time. The design and conduct of the present study were an add-on to the bootcamp given the unique opportunity to gather evidence in a patient-centric environment.

This cross-sectional study included 25 of the 30 bootcamp participants who met the following eligibility criteria: PwHA aged ≥7 years with moderate or severe hemophilia A who were receiving episodic or prophylactic treatment with any of the locally approved therapies for hemophilia A, with or without a history of inhibitors. Individuals with mild hemophilia A, other types of coagulopathy (e.g., hemophilia B, von Willebrand disease) and those with concurrent high-responding inhibitors with or without bleeding control were excluded; the latter due to the high risk of bleeding events, which did not allow them to participate in the bootcamp.

Written informed consent was obtained from participants aged 18 years or older and from parents or legally acceptable representatives if participants were <18 years. Additionally, informed assent was obtained from participants <18 years. An independent ethics committee approved the study protocol (*Fundacion BIOS*, *Bogotá*, *Colombia*).

## Data collection

At different points during the 3-day bootcamp, the following information was collected:

- *PROBE questionnaire*: The questionnaire consists of a total of 29 questions covering four major sections (Demographic, General health problems, Hemophilia-related health problems, and HRQoL using the EQ-5D); it is available in Spanish and is validated internationally [18] and in Colombia [19]. The EQ-5D comprises two parts: the first is the EQ-5D-5L descriptive system, which uses 5 dimensions for describing health states (Mobility, Usual Activities, Self-care, Pain & Discomfort and Anxiety & Depression), each with 5 levels of problems (no, mild, moderate, severe and extreme problem or unable to do), and the EQ-VAS, which incorporates a visual analog scale (VAS) to capture the respondent's overall assessment of their health on a scale from 0 (worst health imaginable) to 100 (best health imaginable). PROBE representatives in Colombia (*Liga Colombiana de Hemofilia*) were contacted to authorize the use of the paper questionnaire in this study. The bootcamp counselors were present at all times while the participants completed the questionnaire to answer any questions and to ensure completeness of the information. The questionnaire was administered once on Day 2.

- Disease knowledge educational session and survey: On the second day of the bootcamp, an educational session was performed focused on disease overview, bleeding events, treatment of acute bleeding, inhibitors, self-care and healthy habits. A disease knowledge test consisting of fifteen (15) multiple-choice questions was developed to address these topics and was administered before and after the training. After the training was conducted and the

questionnaires collected, the patient organization went through each question, explained the correct answers and clarified any doubts from the participants. The survey was administered on Day 2 and repeated on Day 3.

- Patient interviews: Semistructured interviews were performed by trained qualitative researchers in the local language. A three-hour focus group was led by a moderator using an interview guide with questions focused on the patient's health beliefs regarding hemophilia A, understanding of treatment challenges and impacts, and interactions with the health care system. The moderator explored issues both at the individual level and by encouraging discussions among participants. The interviews and focus groups were conducted on Day 2.

### Data analysis

Data analyses were of a descriptive nature, and this study was not designed to test any hypotheses. Hence, a nonprobabilistic convenience sampling method was selected.

The results of the PROBE questionnaire were summarized by absolute and relative frequencies for categorical variables and by medians and interquartile ranges for continuous variables. In the disease knowledge survey, each of the 15 multiple-choice questions was scored (false: 0, true: 1), and the median of the overall results across individuals was obtained. The precision of the median was measured with 95% confidence intervals.

For the qualitative analysis, all interviews were audio recorded, transcribed verbatim and anonymized. Recordings were reviewed to identify overarching themes, and saturation was assessed by comparing concepts elicited in the transcripts and judged to be achieved if no new relevant concepts emerged in the interviews. IPSOS Napoleon Franco S.A. was responsible for the analysis and compilation of the qualitative data from the patient interviews. Statistical analyses were conducted using STATA version 15.0 (quantitative data) and in QDA Miner (qualitative data).

## Results

### Study participants

Twenty-four participants had severe hemophilia A (96%), and the remaining patients had moderate hemophilia A. The median age was 20 years (IQR, 13–30 years), and most participants came from urban areas (60%; n = 15), with the main cities being Medellin, Cali, Bucaramanga and Villavicencio; the remaining participants (40%; n = 10) were from suburban/rural areas in the Antioquia and Santander regions. The median weight was 62 kg (IQR: 51–72 kg), and the median number of years of schooling was 10 (IQR: 6.75–11.25 years) (Table 1). Five individuals participated in the bootcamp but were not eligible for this observational study: two with hemophilia B, two with mild hemophilia A and one with von Willebrand disease.

### PROBE results

In the General health problems section of the PROBE questionnaire, 16 participants (64%) reported health problems during the last 12 months, the most common being hemarthrosis in different joints (44%). Eight participants (32%) required mobility assistance devices, and 22 (88%) reported the use of any type of pain relief medication in the last 12 months. Furthermore, 80% experienced acute pain and 44% experienced chronic pain during the same period. Similarly, 48% reported having problems with activities of daily living (ADLs) related to bathing and using stairs (Table 1).

**Table 1. Demographic data and general health problems reported in the PROBE questionnaire (N = 25).**

| SECTION I Demographic data | | |
|---|---|---|
| *Characteristic* | | *Median, IQR* |
| Age (yrs) | | 20, 13–30 |
| Weight (kg) | | 62, 51–72 |
| Years of school/education completed, yrs | | 10, 6.75–11.25 |
| *Characteristic* | | *n(%) N = 25* |
| Sex, male | | 25 (100) |
| **SECTION II General health problems** | | |
| *Characteristic* | | *n(%) N = 25* |
| Health problems in the last 12 mo | | 16 (64) |
| Description of problem | Hemarthrosis | 11 (44) |
| | Tooth Disease | 1 (4) |
| | Epilepsy | 1 (4) |
| | Join Surgery | 1 (4) |
| | Joint Pain | 1 (4) |
| | Hematoma | 1 (4) |
| Use of a mobility aid or assistive device in the past 12 mo | | 8 (32) |
| Type of aid or device | Compression bandage/wrap | 4 (16) |
| | Orthopedic brace | 2 (8) |
| | Sling | 0 |
| | Cane | 4 (16) |
| | Crutches | 3 (12) |
| | Walker | 1 (4) |
| | Manual wheelchair | 2 (8) |
| | Motorized wheelchair | 0 |
| | Electric scooter | 0 |
| Use of any medication for pain in the past 12 mo | | 22 (88) |
| Frequency of use | Never (0% of the time) | 0 |
| | Rarely (1–5% of the time) | 4 (16) |
| | Occasionally (6–25% of the time) | 9 (36) |
| | Sometimes (26%–50% of the time) | 4 (16) |
| | Frequently (51%–75% of the time) | 5 (20) |
| | Very frequently (76%–99% of the time) | 0 |
| | All of the time (100% of the time) | 0 |
| Acute pain in the past 12 mo | | 20 (80) |
| When does acute pain occur? | Walking | 8 (32) |
| | Climbing stairs | 4 (16) |
| | Nighttime (such as waking up/keeping awake) | 12 (48) |
| | Resting | 5 (20) |
| | Weight bearing | 5 (20) |
| | Other | 6 (24) |

(*Continued*)

**Table 1.** (Continued)

| SECTION I Demographic data | | |
|---|---|---|
| Does your acute pain interfere with any of the following? | General activity | 10 (40) |
| | Mood | 6 (24) |
| | Walking ability | 11 (44) |
| | Normal work (including both work outside the home and housework) | 6 (24) |
| | Attending school | 4 (16) |
| | Relations with others | 3 (12) |
| | Sleep | 7 (28) |
| | Enjoyment of life | 4 (16) |
| | Other | 1 (4) |
| Chronic pain in the past 12 mo | | 11 (44) |
| When does chronic pain occur? | Walking | 7 (28) |
| | Climbing stairs | 6 (24) |
| | Nighttime (such as waking up/keeping awake) | 4 (16) |
| | Resting | 3 (25) |
| | Weight bearing | 1 (4) |
| | Other | 1 (4) |
| Does your chronic pain interfere with any of the following? | General activity | 7 (28) |
| | Mood | 3 (12) |
| | Walking ability | 6 (24) |
| | Normal work (including both work outside the home and housework) | 3 (12) |
| | Attending school | 2 (8) |
| | Relations with others | 2 (8) |
| | Sleep | 1 (4) |
| | Enjoyment of life | 3 (12) |
| Difficulty with ADLs | | 12 (48) |
| Description of problem | Going down stairs | 12 (48) |
| | Going up stairs | 8 (32) |
| | Rising from sitting | 4 (16) |
| | Standing | 6 (24) |
| | Bending to the floor | 4 (16) |
| | Walking on flat surfaces | 4 (16) |
| | Getting in/out of the car | 4 (16) |
| | Going shopping | 6 (24) |
| | Putting on socks | 2 (8) |
| | Lying in bed | 0 |
| | Taking off socks | 2 (8) |
| | Rising from bed | 5 (20) |
| | Getting in/out of the bath | 12 (48) |
| | Sitting | 4 (16) |
| | Getting on/off the toilet | 1 (4) |
| | Heavy domestic duties | 4 (16) |
| | Light domestic duties | 1 (4) |
| | Lifting | 5 (20) |
| | Writing or typing | 1 (4) |
| | Grooming | 0 |
| | Sexual intimacy | 2 (8) |

(*Continued*)

**Table 1.** (Continued)

| SECTION I Demographic data | | |
|---|---|---|
| Current work and/or school life | Working full-time | 5 (20) |
| | Working part-time | 12 (48) |
| | If you are working part-time, is this your personal choice? | 0 |
| | Student full-time | 4 (16) |
| | Student part-time | 11 (44) |
| | On long-term sick or disability leave (more than 6 months) | 0 |
| | Early retirement (before normal retirement age) | 0 |
| | Others (such as unemployment, on parental leave, retired) | 1 (4) |
| Joint surgery or other invasive procedure | | 11 (44) |
| Description of surgery | Aspiration | 2 (8) |
| | Amputation | 0 |
| | Arthroplasty | 2 (8) |
| | Arthrodesis | 0 |
| | Radio or chemical synovectomy | 0 |
| | Surgical synovectomy | 0 |
| | Surgery for removal of a pseudotumor | 1 (4) |
| | Other | 6 (24) |
| *Characteristic* | | *Median, IQR* |
| Days not able to leave your home to go to work, attend school or participate in your normal daily activities due to health in the past 12 months | | 3, 0–12.5 |

IQR: interquartile range; yrs: years; kg: kilograms; mo: months; ADLs: activities of daily living

In the Hemophilia-related problems section of the PROBE questionnaire, seven participants (28%) reported being diagnosed with clinically significant inhibitors, and 13 (52%) reported having more than 2 spontaneous bleeding events in the past 12 months (Table 2). Overall, 76% reported having target joints, and 52% reported a limited range of motion on any joint. Treatment was regularly administered at home (72%), with regular prophylaxis reported as the most common scheme with a median dose per infusion of 1,500 IU (IQR: 1,000–2,000 IU) and most common frequency of 3 times per week. A detailed description of the treatment regimens of PwHA can be found in Fig 1.

Fig 2 shows the results of the EQ-5D questionnaire. The mean score reported by the patients was 80 (IQR: 50–100), reporting a worse health state in the EQ-5D-5L dimensions of pain/discomfort (24% moderate and 4% severe problems) and usual activities (16% moderate problems).

## Disease knowledge survey

The preeducational activities survey had a median score of 0.68 (95% CI: 0.49–0.86), and the posteducational activities survey had a median score of 0.83 (95% CI: 0.68–0.98), resulting in a 15% overall difference between the two paired surveys (95% CI: 1.1–29.3) and a 22% (95% CI: 13–38) ratio increase. Across all patients, there was improvement from baseline for two-thirds

**Table 2. Hemophilia-related health problems reported in the PROBE questionnaire (N = 25).**

| SECTION III Hemophilia-related health problems | | |
|---|---|---|
| *Characteristic* | | *n(%)*<br>*N = 25* |
| Type of Hemophilia | Severe (Factor level below 1%) | 24 (96) |
| | Moderate (Factor level of 1–5%) | 1 (4) |
| | Mild (Factor level above 5%) | 0 |
| Clinically significant inhibitors | | 7 (28) |
| Bleeding in the past 12 mo | 0 bleeding events | 7 (28) |
| | 1 bleeding event | 2 (8) |
| | 2–3 bleeding events | 9 (36) |
| | 4–7 bleeding events | 6 (24) |
| | 8–10 bleeding events | 0 |
| | 11–15 bleeding events | 1 (4) |
| | 16–30 bleeding events | 0 |
| | More than 30 bleeding events | 0 |
| Where do you receive your regular treatment? | Home | 18 (72) |
| | Hemophilia treatment center (HTC) | 5 (20) |
| | Emergency room | 1 (4) |
| | Other | 1(4) |
| | No treatment available | 0 |
| Current treatment regimen | Regular prophylaxis | 18 (72) |
| | Intermittent, "periodic" prophylaxis | 5 (20) |
| | Episodic ("on demand") | 1 (4) |
| | Immune tolerance induction (ITI) | 0 |
| | No treatment available | 0 |
| | Other | 1 (4) |
| Treatment frequency | Daily | 0 |
| | Every other day | 0 |
| | 3 times per week | 18 (72) |
| | 2 times per week | 4 (16) |
| | Once per week | 2 (8) |
| | Other (Not described) | 1 (4) |
| Use of extended (prolonged) half-life treatment product | | 1 (4) |
| Target joints | | 19 (76) |
| Which joint? | Left ankle | 3 (12) |
| | Right ankle | 11 (44) |
| | Left elbow | 3 (12) |
| | Right elbow | 2 (8) |
| | Left knee | 2 (8) |
| | Right knee | 5 (20) |
| More than 2 spontaneous bleeding events in these joints in the past 12 mo | | 13 (52) |
| Reduced range of motion of any joint due to hemophilia | | 15 (56) |

(*Continued*)

**Table 2.** (Continued)

| SECTION III Hemophilia-related health problems | | |
|---|---|---|
| *Characteristic* | | *n(%)* *N = 25* |
| Which joint? | Left ankle | 4 (16) |
| | Right ankle | 6 (24) |
| | Left elbow | 7 (28) |
| | Right elbow | 4 (16) |
| | Left knee | 7 (28) |
| | Right knee | 9 (36) |
| | Other | 1 (4) |
| *Characteristic* | | *Median, IQR* |
| Typical dose of factor VIII concentrate used (IUs per infusion) | | 1500, 1000–2000 |

IQR: interquartile range; yrs: years; mo: months; IUs: international units

of the questions after the educational activities. On the other hand, there were two questions for which a decrease was observed in the score from 100% to 96% correct answers after the training. These two questions were related to FVIII treatment of acute bleeding events and to local measures to handle acute bleeding events (Table 3).

## Qualitative results: Participants' experiences with the disease and treatment

In the bootcamp, the participants received training about hemophilia and the care they should have for their condition (Table 4). Regarding bleeding events, they recognized early signs, and most PwHA knew the route of care and how to access emergency services in case of complications. Participants shared that, in their family setting, there was an environment of overprotection that generated discomfort. The assistance provided by third parties, such as patient associations or institutions, was considered necessary. Some participants expressed that living in locations with difficult access, such as rural areas, limited access to medical visits at home or their transfer to specialized care in case of bleeding complications.

Most participants reported carrying out physical activities such as swimming, cycling, and walking and wished they could improve their quality of life and increase the weekly frequency of exercise. For all participants, the hardest part of the day was the morning due to arthropathy-associated pain. All children reported attending school activities and some of them extracurricular activities, as well; young adults pursued university degrees. Many adults carried out work activities independently, and they reported having problems working as employees due to the time required to manage their disease. These participants also reported that they had identified an important generational change in the treatment of hemophilia, as most of them received cryoprecipitates and plasma FVIII as early treatments, and the current generation of drugs offered more benefits in terms of bleeding prevention, reduction of the consequences and impact on HRQoL.

Most participants mentioned that they received their treatment via an intravenous route, commonly describing complaints such as vascular access, requiring multiple punctures, and discomfort due to local hyperkeratosis and ecchymosis. Some PwHA received the treatment at home, and others had to go to their health care institution to receive the infusion or to collect the treatment; only four self-infusing PwHA were identified in the group. Notably, not all the adult participants recognized the brands of their treatment and, rather, identified them by the

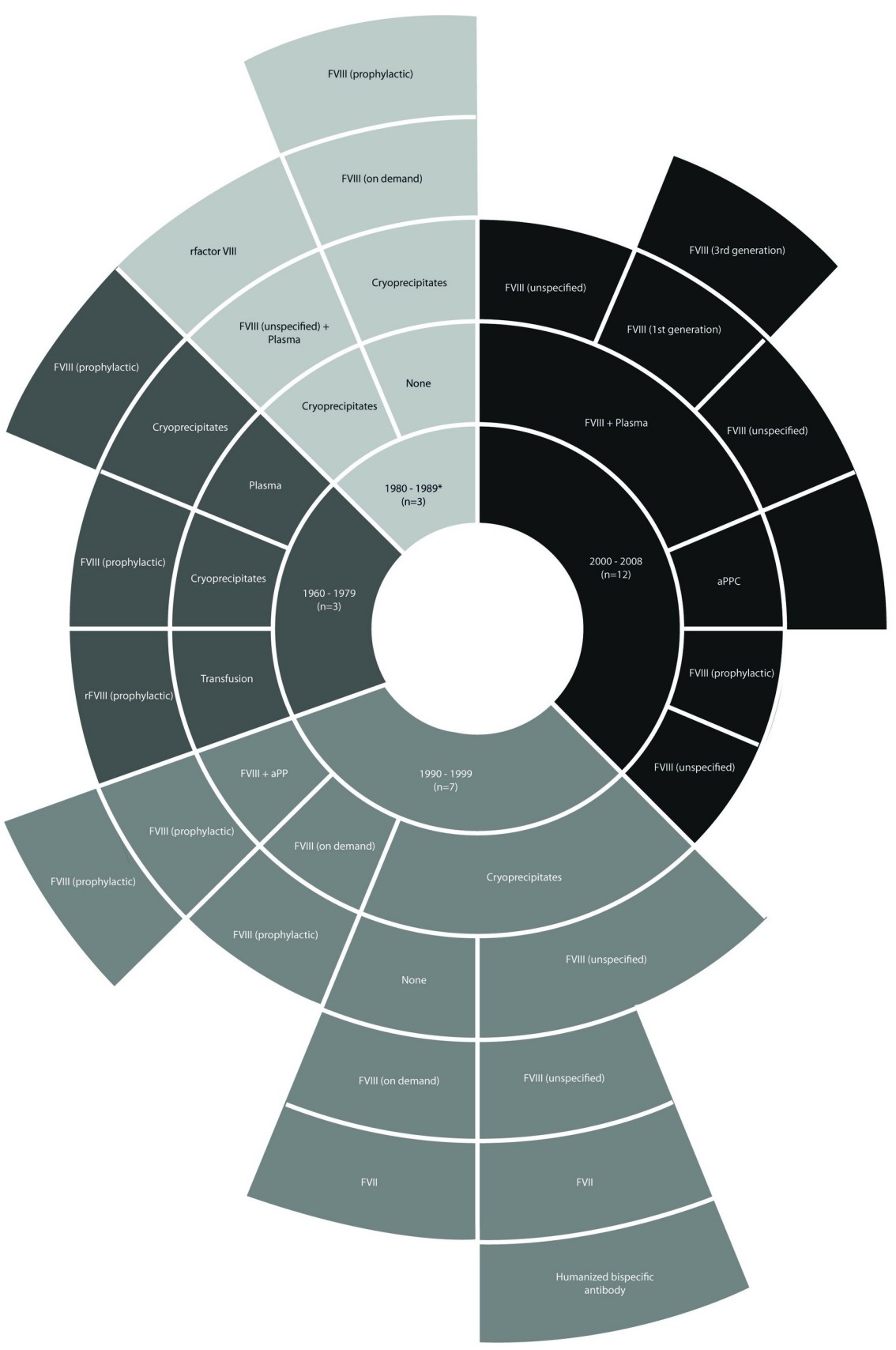

**Fig 1. Treatment regimens among patients with moderate-to-severe hemophilia A (N = 25).** The sunburst plot, which is read from the center outward, depicts the treatment patterns of hemophilia A patients according to the year of birth (grouped by periods). Those patients who were born between 1960 and 1979 received as first-line treatment either blood transfusion, plasma, or cryoprecipitates, with subsequent treatments composed of either cryoprecipitates (if not received as first-line treatment) or FVIII. Similarly, those born between 1980 and 1989 usually started treatment with cryoprecipitates before receiving FVIII on different regimens. Patients born between 1990 and 1999 have a mix of treatment regimens, where FVIII is started as first-line treatment, although more than half of them still received cryoprecipitates as first-line treatment; notably, there were some patients from this decade who were Receiving bridge therapy with factor VII and even treatment with a humanized bispecific antibody. Finally, in the decade between 2000 and 2009, most patients received treatment with FVIII in different schemes (N = 24, *Treatment data not reported by one patient). rFVIII: recombinant factor VIII; FVIII: factor VIII; FVII: factor VII; aPP: activated prothrombin complex concentrate.

name of the marketing authorization holder. Likewise, some were unaware of the infusion scheme they received.

Regarding administrative barriers to treatment, the main issue reported was delivery of the medications after prescription, which could take up to 2 months, followed by difficulties in receiving comprehensive management due to the availability of services. In general, patients were not consulted by their treating physicians regarding a change in treatment, although they were informed in some cases.

Regarding the ideal treatment, patients mentioned that the frequency of administration of the medication was the most important feature. The removal of at least one weekly dose would generate positive changes in their HRQoL, reflected by more freedom in their daily activities, less dependence on third parties, more free time, and fewer venipunctures. When referring to the route of administration, they considered it important to avoid IV, and the most preferred

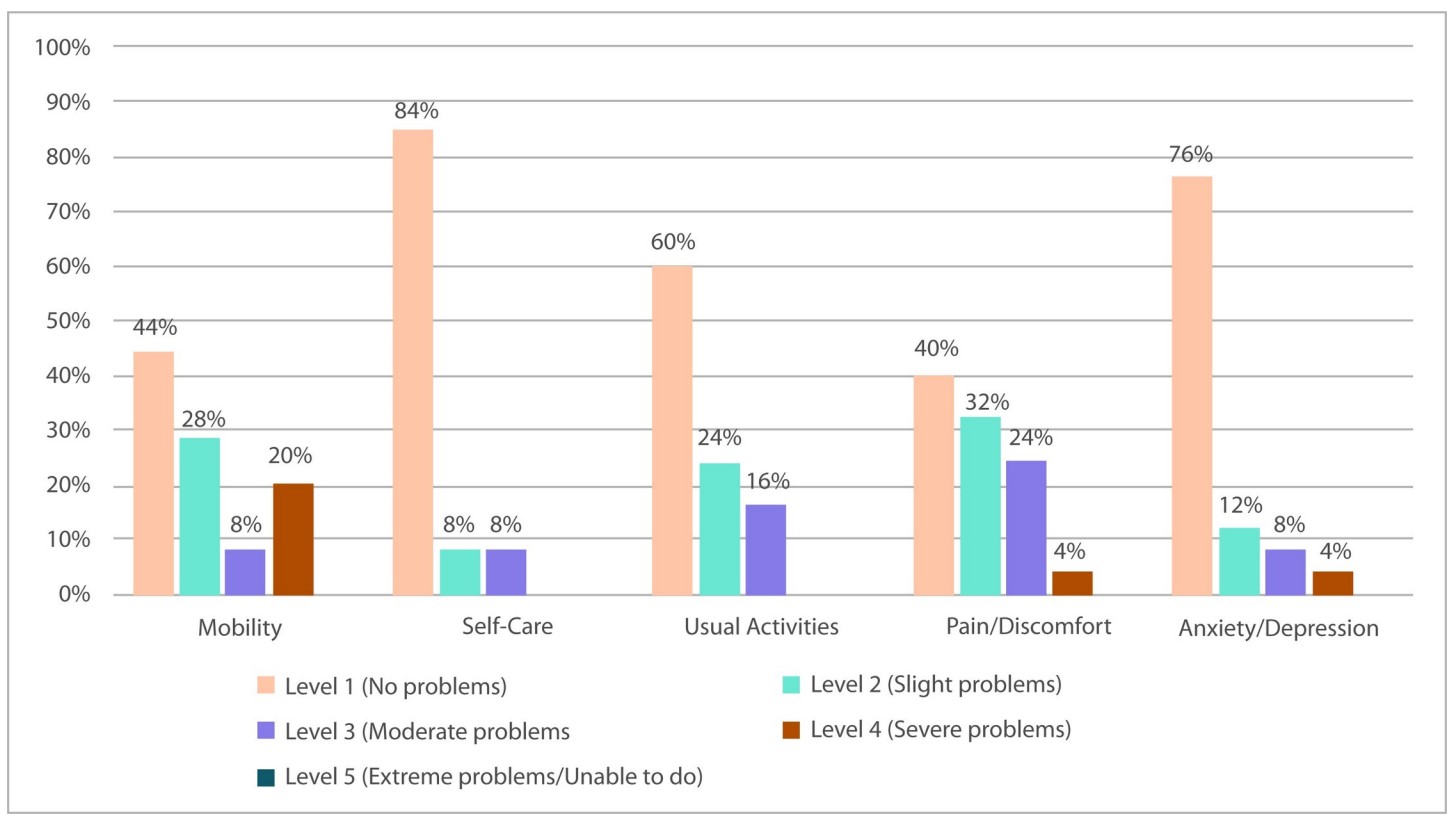

**Fig 2. Proportion of responses by level of severity for EQ-5D-5L dimensions (N = 25).**

**Table 3. Pre- and posteducational activities survey results.**

| Question | Hemophilia A Topic | Score | | Change (%) |
|---|---|---|---|---|
| | | Preeducational activity survey | Posteducational activity survey | |
| 1 | Disease characteristics | 0,92 | 0,96 | 4 |
| 2 | Causes of bleeding | 0,92 | 0,92 | 0 |
| 3 | Severity and differentiation with hemophilia B | 0,76 | 0,92 | 16 |
| 4 | Etiology | 0,64 | 0,96 | 32 |
| 5 | Family history | 0,76 | 0,76 | 0 |
| 6 | Bleeding event overview | 0,6 | 0,92 | 32 |
| 7 | Bleeding event symptoms | 0,56 | 0,76 | 20 |
| 8 | Life-threatening bleeding events | 0,36 | 0,56 | 20 |
| 9 | Inhibitors | 0,68 | 0,88 | 20 |
| 10 | Healthy habits | 0,88 | 0,88 | 0 |
| 11 | Bleeding event treatment with FVIII | 1 | 0,96 | -4 |
| 12 | Local hemostatic measures for bleeding events | 1 | 0,96 | -4 |
| 13 | Management of bleeding events | 0,28 | 0,48 | 20 |
| 14 | Self-care | 0,76 | 0,92 | 16 |
| 15 | Community and advocacy | 0,44 | 0,8 | 36 |
| **Overall Score, Median (95% CI)** | | 0,76 (0.59–0.82) | 0,92 (0.77–0.92) | 16 (6.9–20.8) |

FVIII: factor VIII; CI: confidence interval

route among participants was subcutaneous. Another aspect considered key was that the infusion could be performed at home by the patient or his family (self-infusion). In general, the participants had great expectations from their treatment in terms of frequency and route of administration, and they wanted to learn more about the new drugs available and hoped for the development of oral treatments.

## Discussion

Our study thoroughly described a cohort of PwHA from Colombia, including demographic and clinical characteristics as well as patient-reported outcomes and patient beliefs and experiences related to their understanding of the disease and its treatment. Overall, patients reported an acceptable HRQoL, with worse scores observed in the domains of mobility and pain/discomfort. These results were consistent with those from our qualitative data analysis. PwHA reported experiencing challenges regarding their capacity to act independently daily and their access to care and medicines. Some patients learned to self-infuse prophylaxis treatment because of challenges in reaching a trained health professional who could do so in their living area. As an alternative to the health care system, patients accessed non-health care institutions such as *patient associations* to provide them with legal advice and psychosocial support.

Notably, our qualitative analysis complemented the results of the validated HRQoL and clinical surveys, adding a dimension of desire of the PwHA to act more independently and outside of an overprotective environment, which is known to have a deleterious impact on the management of the disease [20]. Our results showed that this overprotection is still evident and may lead younger PwHA to avoid discussing symptoms or disease knowledge because of their parents' emotional response [9]. The camp was particularly helpful in this aspect, as it provided an opportunity for younger patients to act independently in a supportive environment, which leads to a decreased sense of self-pity and to an encouragement of an independent

**Table 4. Summary of the main insights obtained from semistructured interviews.**

| Category | Variable | Description | Example |
|---|---|---|---|
| Knowledge of the disease and experience | Bleeding events | Route of attention | "Here is a child, it could be anybody, and he fell, his knee inflates, and he waits a moment and rests and applies ice; since he does not improve, he calls the hotline, and they take him in a car. They send him to the health care institution and since he is not doing well, from the health care institution he is transferred to the emergency room, and then, as they treat him in the emergency room, he can be hospitalized or discharged home, but they increase the factor's dose so that it has an effect". |
| | Psychosocial sphere | Feelings | "As we have lived by the norm, being overprotected, sometimes we simply want to be free, and seeking that freedom, we push ourselves in certain activities. Sometimes we also bleed, because we do not apply the factor on time". |
| Living with hemophilia | Activities of daily living | Physical | "I train, I exercise, I do sports, I engage in an activity, and I do sports without limits. We created a group called, "pedaliemos", people with hemophilia hanging out together, riding bicycles". |
| | | Limitations | "It costs me a lot to get off the bike, that is, I almost have to stay in the same position for a while because since I have had my knees flexed the whole time, stretching them costs me a lot due to my arthropathy". |
| Knowledge of the disease and experience | Treatment | Generational comparisons of treatments | "I would have liked to be born in this era where prophylactic treatment prevents joint bleeding and its complications". |
| | | Frequency and place of administration Implications | "Well, it takes 40 minutes. Imagine for one to commute an hour and a half, they do the procedure and then another hour and a half back". |
| | | Infusion | "I learned to place an IV by myself and then they give me my medicine; I have my fridge there and on prophylaxis days, I apply it myself". |
| | | Route of administration and repercussions | "They tried 6 times, and one says let us wait a moment, let us rest, because a child punctured so many times generates anxiety". |
| Barriers to treatment | Administrative | Authorizations Legal procedures | "I started a legal procedure, I put in a complaint to the insurer, before the Secretary of Health, and in other health departments, specifying that they have sent me many therapies, and they have not wanted to approve them". "The public officer of the city accompanied me and demanded that the insurer at that time answer the complaints; he was the director of the city's legal office, a lawyer for the rights of children and minors and another person at that meeting". |
| | Social aspects | Support network | "In my house, I have had to live alone since I was 17; the "league" (patient association) provides legal advice in case we have limitations to health service access, they provide socioeconomic support through donations for low-income people". |
| Ideal Treatment | Medicine | Frequency | "However, to change from applying 3 times to 2 times or once a week, the truth is that's something that one does not believe". "If I removed 1 infusion, I would have more time to do my activities". "One less application would be a benefit and change in my life". |
| | Application | Results | "The ideal medicine would be the one that does not make me bleed spontaneously and that allows me to demand more of myself in my daily activities as an almost normal person, despite the consequences; to be able to demand more of myself, to be able to walk, to be able to go out with my dogs". |

lifestyle [8, 21]. Problems in the working environment because of the disease, especially when employed by others, were also highlighted during the qualitative assessment.

Patients reported current health problems related to their condition, including chronic pain, and more than 80% of them reported acute pain in the last year. Approximately half of the patients reported health issues that impacted their daily activities, which was corroborated by the HRQoL analysis. Seven participants reported a full state of health in the HRQoL survey, and none of the 25 PwHA reported severe problems performing all five activities listed. The median EQ-VAS score was 80 (IQR: 50–100), showing a close to normal result in comparison to that in the general population in Colombia [22]. These results are consistent with those of prior investigations on QoL in similar cohorts of PwHA from Latin America [14–16]. However, our study differs in aspects such as a younger cohort that was more prone to being self-demanding physically and highlighting health concerns in the areas of pain and mobility. One

potential explanation for the overall self-perceived high quality of life despite health-related challenges was the prescription of a prophylactic regimen, which was the most commonly used regimen in our cohort of PwHA [15, 23, 24]. However, this hypothesis remains to be explored in observational settings.

Our study also evaluated the impact of an educational activity on the understanding of the disease and its treatment among PwHA by using a disease knowledge questionnaire (S1 File). As expected, patients showed an appropriate preeducational understanding of the disease, inheritance pattern and severity, as well as the identification and management of bleeding events. After the educational training, and despite the low margin for improvement, we observed an average of a 15% difference in the test score before and after the educational intervention. These results reinforce previous recommendations to act on the patients' understanding and acceptability of their disease and treatment, as there is always room for improvement and enhancing quality of life [25, 26]. Indeed, patients have high expectations for next-generation treatments, especially related to a reduced burden of administration, improved outcomes and more independence to live a "normal life" [27, 28]. The confidence that new products may allow them to increase physical activity was considered notably important by PwHA. In this regard, patient beliefs are of crucial importance for treatment adherence [29], reinforcing the relevance of combined qualitative–quantitative research approaches to understanding patients' journeys and increasing the effectiveness of established and new therapies.

Pain and disability have been reported as the most prevalent problems in large cohort studies among individuals with hemophilia in association or not with chronic hemophilic arthropathy [30–32]. Bleeding control can be defined as fewer than 4 annual spontaneous bleeding events, but PwHA on primary prophylaxis may suffer substantially fewer bleeding events [33]. On the other hand, patients with on-demand treatment can have up to 20–50 bleeding events per year [4, 34]. Our data showed that PwHA on regular prophylaxis who started later in life (>7 yrs) had more than 3 bleeding events per year, a finding that has been previously reported in this population (7.4 bleeding events a year). This fact enhances the need for further individual adaptation of treatment regimens under a new concept of personalized treatment [33]. Although prophylaxis-based therapy was frequently observed in this study (76%), only 4 patients mentioned practicing self-infusion during the focus group. In Colombia, the progress of self-infusion practice among PwHA has been slow, which can be due to the implementation of a home health care model in the country to ensure high adherence among patients receiving prophylaxis, as it has been reported in other countries that discontinuation of prophylaxis is commonly observed among young adults transitioning from childhood, which may be due to transitioning from parent care to self-care [35]. Another reason is the need for close supervision of the appropriate self-infusion technique and correct use of the medicine by patients. For self-infusion to be successful, it requires not only that PwHA be highly motivated and appropriately educated but also that regular check-ups of the quality of the infusion procedure, including reminders for washing hands and checking the medicine information, are performed [36, 37]. This highlights the importance of continuing the focus on patient and caregiver education around self-infusion practices to increase its use in Colombia, as recommended by the World Federation of Hemophilia (WFH) guidelines [38].

Regarding HRQoL, our study showed that self-care and usual activities were the least affected activities, as has been reported in large population-based studies [32]. The more affected domains were pain/discomfort and anxiety/depression, which correlates with the data previously discussed. For PwHA, a reduction in the frequency of and time spent performing the injections (once per week) would generate positive changes in their QoL, reflected in more liberty in their daily activities, less dependence on third parties, more free time and fewer venipunctures.

There are some limitations and strengths that must be considered in interpreting the results of this study. The limitations include the risk for selection bias considering that the PwHA were recruited from those who participated in the camp and the small sample size. However, hemophilia A is a rare disease that limited participation in this study, and the population included represented PwHA from urban and rural areas in an LMIC. Furthermore, the statistical analysis was explorative for descriptive purposes and not intended for hypothesis testing or generalization to other areas. In addition, due to data privacy restrictions, we were not able to match at the patient level the information from the PROBE questionnaire with the answers collected in the focus group. The strengths of this study include the combined quantitative and qualitative approach that provided a holistic vision of the health care experiences of PwHA from an LMIC regarding their disease and treatment. In addition, from the public health perspective, this study provides further evidence of the importance of educating patients as a strategy to improve the comprehensive management of their disease. Despite the use of a convenience sampling method, our cohort of 25 PwHA showed demographic characteristics like those reported in the nationwide registry of hemophilia A patients, thus reinforcing its generalizability [13]. Finally, HA is a rare disease; thus, observational studies using primary data collection from PwHA tend to be few. On the other hand, our study had strong internal validity allowed by the completeness of data collection, the mixed qualitative–quantitative approach, and the prespecified analysis plan. Therefore, we believe that our results add relevant information to the bulk of evidence already available, hence increasing the knowledge base.

## Conclusion

The value of involving PwHA and their perspectives in the decision-making process of their disease and treatment has gained increasing relevance in recent years. This study explored the perspectives of PwHA regarding their disease and treatment experience in the context of a disease-specific educational bootcamp and provided unique insights into their disease knowledge, QoL and expected treatment outcomes. The outcomes of this study could serve as key complementary information to health systems and policy-makers to implement a patient-centered framework for public health decision-making that includes a comprehensive analysis of the value and impact of novel hemophilia therapies such as EHL FVIII products, which have the potential to decrease the treatment burden and improve adherence, resulting in better health outcomes.

## Supporting information

**S1 File. Disease knowledge survey.**
(PDF)

**S2 File. Study's minimal data set.**
(XLSX)

## Acknowledgments

The study team wish to extend a special thanks to the "Liga Antioqueña de Hemofilia" for their support in coordinating and conducting the Hemophilia Patient Bootcamp, and to the study participants who provided valuable insight into their experience with the disease and its treatment. The authors would also like to acknowledge the support of Dr. Mark Skinner and the PROBE study for the review of the manuscript.

## Author Contributions

**Conceptualization:** Liliana Torres, Oscar Peñuela, Maria del Rosario Forero, Marcela Rivera, David Vizcaya, Juan-Sebastian Franco.

**Data curation:** Juan Satizabal, Ximena Salazar, Diana Benavides, Raul Gamarra.

**Formal analysis:** Liliana Torres, Oscar Peñuela, Juan Satizabal, Ximena Salazar, Diana Benavides, Raul Gamarra, David Vizcaya, Juan-Sebastian Franco.

**Funding acquisition:** Juan-Sebastian Franco.

**Methodology:** Liliana Torres, Oscar Peñuela, Maria del Rosario Forero, Marcela Rivera, David Vizcaya, Juan-Sebastian Franco.

**Project administration:** Liliana Torres, Oscar Peñuela, Maria del Rosario Forero, Juan-Sebastian Franco.

**Supervision:** David Vizcaya, Juan-Sebastian Franco.

**Validation:** Liliana Torres, Oscar Peñuela, Maria del Rosario Forero, Juan Satizabal, Ximena Salazar, Diana Benavides, Raul Gamarra, Marcela Rivera, David Vizcaya, Juan-Sebastian Franco.

**Writing – original draft:** Liliana Torres, David Vizcaya.

**Writing – review & editing:** Liliana Torres, Oscar Peñuela, Maria del Rosario Forero, Juan Satizabal, Ximena Salazar, Diana Benavides, Raul Gamarra, Marcela Rivera, David Vizcaya, Juan-Sebastian Franco.

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
