## [Decision Letter · Decision Letter 0]

17 Apr 2023

PONE-D-22-24662Quality of Life, self-reported outcomes and impact of education in persons with moderate and severe Hemophilia A: an integrated perspective from a Latin American countryPLOS ONE

Dear authors,

Thank you for submitting your manuscript to PLOS ONE. After careful consideration, we feel that it has merit but does not fully meet PLOS ONE’s publication criteria as it currently stands. Therefore, we invite you to submit a revised version of the manuscript that addresses the points raised during the review process.

We look forward to receiving your revised manuscript.

Kind regards,

Wolfgang Miesbach, MD

Academic Editor

PLOS ONE

“Yes, this study was sponsored by Bayer S.A. Colombia.”

“This study and manuscript development was fully funded by Bayer. The study team wish to extend a special thanks to the “Liga Antioqueña de Hemofilia” for their support in coordinating and conducting the Hemophilia Patient Bootcamp, and to the study participants who provided valuable insight into their experience with the disease and its treatment. The authors would also like to acknowledge the support of Dr. Mark Skinner and the PROBE study for the review of the manuscript. We thank Dr. Ana Maria Perez who provided english editing and review.”

“Yes, this study was sponsored by Bayer S.A. Colombia.”

“Juan Satizabal, Ximena Salazar, Diana Benavides and Raul Gamarra are employees of IPSOS Napoleon Franco S.A., the outcomes research consultancy commissioned by Bayer to conduct the qualitative analysis of this study. Liliana Torres, Oscar Peñuela, Maria del Rosario Forero, Marcela Rivera, David Vizcaya and Juan-Sebastian Franco are employees of Bayer. All authors have no further conflicts to disclose.”

Additional Editor Comments:

This is a well written and important study. Please see below the comments to be addressed. Please answer them very carefully, particularly the topic of self-infusion.

Reviewers' comments:

Reviewer's Responses to Questions

**Comments to the Author**

1. Is the manuscript technically sound, and do the data support the conclusions?

Reviewer #1: Yes

2. Has the statistical analysis been performed appropriately and rigorously? 

Reviewer #1: N/A

3. Have the authors made all data underlying the findings in their manuscript fully available?

Reviewer #1: Yes

4. Is the manuscript presented in an intelligible fashion and written in standard English?

Reviewer #1: Yes

5. Review Comments to the Author

Reviewer #1: Thank you for submitting this manuscript for consideration, it offers an important insight into the care and treatment of people with haemophilia in South America.

There are a couple of issues that it would be helpful to resolve before the manuscript is published.

1. The English is very good, but there are occasions where the sentence construction isn't quite right. There are also a small but frequent number of text errors which should be addressed. It would benefit the manuscript for you to have it reviewed by a language/copy editing service if possible.

2. You have mentioned the relatively small number of participants who were able to self infuse but have not discussed this further. It would be interesting and would benefit the paper if you were to discuss this issue further, including why there are so few. what are the barriers, and how self infusion rates might be improved. It would also be interesting to see if there were any differences between the QoL of those who were able to self infuse, those who were on Prophylaxis administered by others and those who were on on demand etc.

6. PLOS authors have the option to publish the peer review history of their article (what does this mean?). If published, this will include your full peer review and any attached files.

Reviewer #1: No

---

## [Author Response · Author response to Decision Letter 0]

28 May 2023

Doctor

Wolfgang Miesbach

Academic Editor

PLOS ONE

Dear Dr. Miesbach,

We greatly appreciate the opportunity to further revise our manuscript entitled “Quantitative and qualitative exploration of the Quality of Life, self-reported outcomes and impact of education among people with moderate and severe hemophilia A: An integrated perspective from a Latin American country”. The thoughtful comments and kind suggestions provided by you and the reviewer have helped in positioning this manuscript. We addressed the reviewers’ comments point-by-point.

The manuscript was modified to comply with the journal’s requirements, including style, modification of Acknowledgements section and provision of study's minimal data set as Supporting Information. As indicated, we would like to kindly ask for your support updating the following statements on the online submission:

- Role of Funder statement: “The study was funded by Bayer SA Colombia. Medical writing was provided by Ana Maria Perez and editorial assistance was provided by American Journal Experts, North Carolina, USA, funded by Bayer SA Colombia. This study was conducted in accordance with Good Pharmacoepidemiology Practices. The study protocol was approved by an expert committee from Bayer prior to study initiation and the analysis dataset is available as supplementary material.”

- Competing Interests statement: “Juan Satizabal, Ximena Salazar, Diana Benavides and Raul Gamarra are employees of IPSOS Napoleon Franco S.A. Liliana Torres, Oscar Peñuela, Maria del Rosario Forero, Marcela Rivera, David Vizcaya and Juan-Sebastian Franco are employees of Bayer. All authors have no further conflicts to disclose. This does not alter our adherence to PLOS ONE policies on sharing data and materials.”

In addition, the Discussion was revised to address the low self-infusion rate observed in the study and the paper went through an editing service provider, according to your suggestions. 

Response to Reviewers

Reviewer #1 Comments: 

We are grateful for the expert comments and excellent advice we have received. In responding your respective comments, we used abbreviation “A” in place of “authors”. The point-by-point responses to the comments are as follows:

1. The English is very good, but there are occasions where the sentence construction isn't quite right. There are also a small but frequent number of text errors which should be addressed. It would benefit the manuscript for you to have it reviewed by a language/copy editing service if possible.

A/ The manuscript was reviewed by an editing service (American Journal Experts) according to the reviewer suggestions.

2. You have mentioned the relatively small number of participants who were able to self infuse but have not discussed this further. It would be interesting and would benefit the paper if you were to discuss this issue further, including why there are so few. what are the barriers, and how self infusion rates might be improved. It would also be interesting to see if there were any differences between the QoL of those who were able to self infuse, those who were on Prophylaxis administered by others and those who were on demand etc.

A/ The Discussion was reviewed to address the low self-infusion rates observed in the study in the context of the Colombian health system and how education is key to increase the rates among PwHA in our country, with the inclusion of additional references. 

Regarding the differences between the QoL among those who self infuse, this information is not available since it was captured in the qualitative portion of the study through the focus group and due to data privacy reasons, we’re not able to match the EQ5D responses with the individual responders from the focus group. Furthermore, the PROBE questionnaire does not specifically ask about self-infusion (only up to the primary treatment regime and where do the patients receive such treatment). Therefore, we are not able to perform this analysis with the quantitative information that is available. We have clarified this in the Discussion as a study limitation

---

## [Editor Report · Decision Letter 1]

19 Jun 2023

Quality of life, self-reported outcomes and impact of education among people with moderate and severe hemophilia A: An integrated perspective from a Latin American country

PONE-D-22-24662R1

Dear Dr. Franco,

We’re pleased to inform you that your manuscript has been judged scientifically suitable for publication and will be formally accepted for publication once it meets all outstanding technical requirements.

Kind regards,

Wolfgang Miesbach, MD

Academic Editor

PLOS ONE

---

## [Editor Report · Acceptance letter]

26 Jun 2023

PONE-D-22-24662R1 

Quality of life, self-reported outcomes and impact of education among people with moderate and severe hemophilia A: An integrated perspective from a Latin American country 

Dear Dr. Franco:

I'm pleased to inform you that your manuscript has been deemed suitable for publication in PLOS ONE. Congratulations! Your manuscript is now with our production department. 

Kind regards, 

on behalf of

Dr. Wolfgang Miesbach 

Academic Editor

PLOS ONE